# Peer review of "Knowledge, attitude, and practice toward hyperuricemia among healthcare workers in Shandong, China"

_PeerJ, doi:10.7717/peerj.17926_

## Round 0.1 · original submission · Major Revisions

Dear Dr. Gao,

If you feel you can revise your manuscript appropriately in response to the reviewers' comments, please revise your manuscript and send us your revised manuscript, including your response to the reviewers.

Prof. Yoshinori Marunaka, MD, PhD
Academic Editor
PeerJ Life & Environment

·

Basic reporting

Reviewing the knowledge, attitude, and practices of healthcare workers regarding hyperuricemia in Shandong, China holds pivotal significance as it sheds light on potential gaps in awareness, treatment protocols, and preventive measures. Such insights can inform targeted interventions, ultimately improving patient outcomes and public health in the region. Here are some suggestions I have for improving this paper.
Introduction: In this section, discuss the prevalence of hyperuricemia, both globally and specifically in China, highlighting any regional variations or trends. Emphasize the burden of hyperuricemia on healthcare systems and individuals, including healthcare workers themselves.
Briefly mention the current treatment options available for hyperuricemia, such as lifestyle modifications, pharmacological interventions (e.g., urate-lowering drugs), and dietary recommendations. This addition would help provide readers with a broader perspective on the management strategies for hyperuricemia and reinforce the significance of the study in evaluating current practices among healthcare workers.
A recent work should be cited for lines 39–40. You could use the article below as a reference.
Ashiq K, Bajwa MA, Tanveer S, Qayyum M, Ashiq S, Khokhar R, Abid F. A comprehensive review on gout: The epidemiological trends, pathophysiology, clinical presentation, diagnosis and treatment. JPMA. The Journal of the Pakistan Medical Association. 2021 Apr 1;71(4):1234-8.

Discussion: Discuss the practical implications of the study findings for healthcare practice and policy in Shandong, China. Consider how improving knowledge, attitudes, and practices among healthcare workers could lead to better prevention, diagnosis, and management of hyperuricemia in the region. Propose potential interventions or strategies based on the study findings to address gaps in knowledge, attitudes, and practices among healthcare workers. This could include educational programs, training initiatives, or policy changes aimed at enhancing awareness and improving patient care.
Consider the generalizability of the study findings to other settings or populations beyond Shandong, China. Discuss any factors that may limit the generalizability of the findings and their implications for future research.

Experimental design

The study employed a cross-sectional design to assess the knowledge, attitudes, and practices (KAP) regarding hyperuricemia among healthcare workers in Shandong, China. Ensuring clarity and precision in this section is crucial for readers to understand the study's design and methodology accurately. I would like to recommend making the few corrections listed below.
Detail the process of participant recruitment, including how invitations to participate were sent out via the online platform, any inclusion or exclusion criteria, and how informed consent was obtained.
Additionally, it should be noted that the study concentrated exclusively on physicians and nurses, with pharmacists omitted from the participant pool. Although pharmacists are integral to healthcare delivery, their inclusion was not pursued in this study. Could you please provide more context or specific reasons why pharmacists were not considered for the study? This will help in formulating a statement that accurately reflects the rationale behind their exclusion.
Describe the sampling method used to recruit participants, whether it's random sampling, convenience sampling, or another approach.
Describe the instructions provided to participants, the timeframe for completing the forms, and any reminders or follow-ups sent to non-respondents.

Validity of the findings

Acknowledge potential limitations of the study, such as selection bias, response bias, or limitations inherent to the cross-sectional design. During the statistical analysis, was there any bias created? If yes, how it was managed?

·

Basic reporting

Thank you for inviting me to review the paper entitled "Knowledge, attitude, and practice toward hyperuricemia among healthcare workers in Shandong, China." I have gone through the manuscript, generally, the research question is clear, the objectives research hypothesis are well-settled, and the findings are significant, well-presented, and discussed. However, for further improvement of this paper, I have some comments and suggestions that need to be addressed by the authors as follows:

Experimental design

Introduction section:
-In lines 48, 49 the author stated that "Gout and asymptomatic hyperuricemia are often treated with urate-lowering therapy (ULT)"
Treating asymptomatic hyperuricemia is a matter of controversy as the latest American College of Rheumatology (ACR) guidelines (2023) advise against routine treatment of asymptomatic hyperuricemia in the absence of related symptoms or conditions. Therefore, the authors need to modify the above-mentioned statement.
(FitzGerald JD, et al 2020 American College of Rheumatology Guideline for the Management of Gout. Arthritis Care Res (Hoboken). 2020 Jun;72(6):744-760. doi: 10.1002/acr.24180. Epub 2020 May 11. Erratum in: Arthritis Care Res (Hoboken). 2020 Aug;72(8):1187. Erratum in: Arthritis Care Res (Hoboken). 2021 Mar;73(3):458.)

- Also, in lines 55-57 the authors mention that "accumulation of urate crystals, elevated SUA levels have been found to cause silent tissue 56 damage, leading to several diseases such as hypertension, dyslipidemia, obesity, metabolic 57 syndrome, type 2 diabetes, cardiovascular disease, and chronic kidney disease (CKD)"
The causal relationship between asymptomatic hyperuricemia and the development of these diseases is not fully established. It's unclear whether elevated SUA is a direct cause or just a marker of underlying metabolic disturbances.
The 2023 ACR guidelines, as discussed above, do not recommend routine treatment of asymptomatic hyperuricemia due to the lack of strong evidence that it prevents the development of these diseases. Hence, the above-mentioned statement needs to be modified.

Materials and Methods:
-Can you please specify how did you calculated the study sample? Also, no mention of the total study population and the sampling technique.
-Was the questionnaire used in this study a validated one, if so, please provide an appropriate citation for that.
-There is no information about missing data and how the author handled the missing data.

Validity of the findings

Result:
- In lines 129-130, reveal a very surprising finding, nephrologists have low knowledge scores. What is the reason for this?
- Additionally, it is notable that no rheumatologists were interviewed. Why were excluded when their expertise in treating severe gout cases, whereby their attitudes and knowledge on hyperuricemia is important?
- In lines 142-143: There is a lack of a question about the side effect of urate-lowering therapy Allopurinol, especially given the ethnic predisposition of Chinese populations to getting SJS due to Allopurinol.

Discussion:
- No comparison is made between this study and previous similar studies in the same country (ie China).
- In lines 179-181: The comparison may not be entirely fair given PCP in Saudi Arabia are mainly staffed by non-training physicians which may influence knowledge.
Conclusion:
-Consider adding a sentence or two on the practical implications of the findings - how can these findings enhance hyperuricemia management in the healthcare setting?
-Briefly mention potential future research directions to build on the current findings.

Reviewer 3 ·

Basic reporting

Gao and coauthors presented an interesting study: Knowledge, Attitude, and Practice Toward Hyperuricemia Among Healthcare Workers in Shandong.
The authors have undertaken a comprehensive approach, addressing an interesting and important topic, presenting a cross-sectional observation study, and discussing some previously published studies, all of which lend credibility to their presented conclusions.
However, the research does not provide significant novelties that would significantly impact current and future clinical practice.

Experimental design

no comment

Validity of the findings

no comment

Additional comments

The figures should be explained to potential readers in more detail.

---

## Round 0.2 · accepted · Accept

Dear Dr. Gao,

Congratulations again, and thank you for your submission.

Warm regards,
Yoshi
Prof. Yoshinori Marunaka, M.D., Ph.D.
Academic Editor
PeerJ Life & Environment

·

Basic reporting

Satisfactory. All comments have been addressed; I have no further comments.

Experimental design

Satisfactory. All comments have been addressed; I have no further comments.

Validity of the findings

Satisfactory. All comments have been addressed; I have no further comments.